


# Projections of streamflow intermittence under climate change in European drying river networks

Louise Mimeau[1,*], Annika Künne[2,3,*], Alexandre Devers[1,4], Flora Branger[1], Sven Kralisch[3], Claire Lauvernet[1], Jean-Philippe Vidal[1], Núria Bonada[5,6], Zoltán Csabai[7,8,9], Heikki Mykrä[10], Petr Pařil[11], Luka Polović[11,12], and Thibault Datry[1]

[1]UR RiverLy, INRAE, Villeurbanne, France
[2]Institute of Geography, Friedrich Schiller University Jena, Jena, Germany
[3]Flood Information Center, State Office for Environment, Mining and Nature Conservation, Jena, Germany
[4]EDF-DTG, Saint-Martin-Le-Vinoux, France
[5]FEHM-Lab (Freshwater Ecology, Hydrology and Management), Departament de Biologia Evolutiva, Ecologia i Ciències Ambientals, Facultat de Biologia, Universitat de Barcelona (UB), Diagonal 643, 08028 Barcelona, Catalonia/Spain
[6]Institut de Recerca de la Biodiversitat (IRBio), Universitat de Barcelona (UB), Diagonal 643, 08028 Barcelona, Catalonia/Spain
[7]Department of Hydrobiology, University of Pécs, Pécs, Hungary
[8]HUN-REN Balaton Limnological Research Institute, Tihany, Hungary
[9]HUN-REN Centre for Ecological Research, Institute of Aquatic Ecology, Debrecen, Hungary
[10]Finnish Environment Institute, Nature Solutions, P.O. Box 413, FI-90014, Oulu, Finland
[11]Department of Botany and Zoology, Faculty of Sciences, Masaryk University, Brno, Czechia
[12]University of Zagreb, Faculty of Science, Department of Biology, Zagreb, Croatia

**Correspondence:** Louise Mimeau (louise.mimeau@inrae.fr)

**Abstract.** Climate and land use changes, as well as human water use and flow alterations, are causing worldwide shifts in river flow dynamics. During the last decades, low-flows, flow intermittence, and drying have increased in many regions of the world, including Europe. This trend is projected to continue and exacerbate in the future, resulting in more frequent and intense hydrological droughts. However, due to a lack of data and studies on temporary rivers in the past, little is known about the

processes governing the development of flow intermittence and drying, their timing and frequency, as well as their long-term evolution under climate change. Moreover, understanding the impact of climate change on the drying up of rivers is crucial to assess the impact of climate change on aquatic ecosystems, biodiversity and functional integrity of freshwater systems.

This study is one of the first to present future projections of drying in intermittent river networks, and to analyze future changes in the drying patterns at high resolution spatial and temporal scale. The flow intermittence projections were produced

using a hybrid hydrological model forced with climate projection data from 1985 until 2100 under three climate scenarios in six European drying river networks. The watersheds areas are situated in different biogeographic regions, located in Spain, France, Croatia, Hungary, Czechia, and Finland, and their areas range from 150 km$^2$ to 350 km$^2$. Additionally, flow intermittence indicators were developed and calculated to assess changes in the characteristics of the drying spells at the reach scale, and changes in the spatial extent of drying in the river network at various time intervals.

The results show that drying patterns are projected to increase and expand in time and space in all three climate scenarios, despite differences in the amplitude of changes. Temporally, in addition to the average frequency of drying events, the dura-





tion also increases over the year. Seasonal changes are expected to result in an earlier onset and longer persistence of drying throughout the year. Summer drying maxima are likely to shift to earlier in the spring, with extended drying periods or additional maxima occurring in autumn, and in some regions extending into the winter season. A trend analysis of extreme events

shows that the extreme dry spells observed in recent years could become regular by the end of the century. Additionally, we observe transitions from perennial to intermittent reaches in the future.

## 1 Introduction

Low flow situations and flow intermittence, where sections of rivers run dry, are becoming increasingly common around the world due to climate change and human water use or flow alterations. These factors do often interact in complex ways and

can lead to prolonged and intensified periods of dryness and drought, disrupting river ecosystem functions and threatening the valuable services they provide (Pal et al., 2015; Datry et al., 2018, 2022; Van Loon et al., 2016). Currently, more than half of the length of global river networks are alternating between flowing and dry conditions, i.e. 51-60% of the 64 million kilometers of river and stream channels, particularly in arid and semi-arid regions (Messager et al., 2021). These intermittent rivers are characterized by the cessation of streamflow, where extreme low flow can result in pool formation or in fully dry, terrestrial

riverbeds (Datry, 2017). This expansion and contraction of temporal river networks can be caused by various conditions of pedo-lithological, topographical, and meteorological origin (Van Meerveld et al., 2019). In addition to these natural causes, anthropogenic drivers, such as water abstractions or land use change, can also trigger flow intermittence or increase the duration and frequency of dry spells (Datry et al., 2022; Van Loon et al., 2016). Observed and simulated hydrological processes and river flow alterations show an increase of low flow situations and drying during the last decades in many river basins around

the world (Gudmundsson et al., 2021; Gelfan et al., 2023).

Changes in natural flow regimes can impact freshwater ecology, leading to habitat degradation, loss of species adapted to specific flow conditions, reduced biodiversity and a decline in overall ecosystem health (Poff et al., 1997). Even though rivers cover less than 2% of the Earth's surface, they are home to around 13% of all known species, making them a hotspot for biodiversity (Strayer and Dudgeon, 2010; Datry et al., 2023). Intermittent rivers and streams, in particular, support species

that have evolved to adapt to these variable conditions of flow dynamics, thereby forming unique ecosystems (Datry et al., 2014). Increases of hydrological droughts and shifts from perennial to intermittent flow regimes threaten both aquatic and terrestrial ecosystems by challenging species adapted to intermittent flow regimes (Leigh et al., 2016; Pörtner et al., 2021), and hence compromising river biodiversity as well as ecosystem functions and services (Tonkin et al., 2019; Bond et al., 2008; Steward et al., 2012; Tramblay et al., 2021; Gudmundsson and Seneviratne, 2016). Besides, recent research has highlighted the

significant role of non-perennial rivers in contributing to annual greenhouse gas emissions (GHG), emphasising the importance of including non-perennial rivers in global GHG estimates to accurately reflect the full contribution of river networks to the carbon cycle (López-Rojo et al., 2024; Silverthorn et al., 2024). A further aspect that has recently been studied by Barthelemy et al. (2024) suggest that flow intermittence also contributes to increasing plastic fragmentation in river ecosystems.





Projections of the Intergovernmental Panel on Climate Change (IPCC) indicate that the persistence of global warming
throughout the 21st century will be largely determined by the socio-economic pathways chosen by societies (IPCC, 2021),
ranging from sustainable development to scenarios dominated by fossil-fueled growth. The Shared Socioeconomic Pathways
(SSPs) framework represents these diverse scenarios, providing a detailed foundation for understanding how socio-economic
decisions might impact future climate conditions, as well as climate change mitigation and adaptation strategies (Riahi et al.,
2017). However, the continuation of global warming will lead to increased evapotranspiration, mainly driven by higher temper-
atures and changes in atmospheric moisture demand, which exacerbate the frequency and intensity of dry spells and droughts
(Naumann et al., 2018; Calvin et al., 2023). Depending on the environmental characteristics of each river basin, the projected
impacts of climate change on river runoff are manifold, ranging from increased occurrence of highflows to an amplification of
low flow situations (Gusev et al., 2018). The increase of extreme streamflow patterns under climate change was also observed
in various river basins located in different European climate zones (Schneider et al., 2013). Studies indicate that the frequency
and severity of low flow conditions, streamflow droughts as well as drying events are projected to increase significantly until
the end of the century in Europe (Feyen and Dankers, 2009), potentially causing shifts from perennial to intermittent flow
regimes (Döll and Schmied, 2012; Sarremejane et al., 2022).

Despite recent advancements, intermittent rivers and ephemeral streams remain understudied, leaving critical gaps in our
understanding of ecohydrological interactions across different spatial and temporal scales, especially as climate change and
human water use increasingly threaten ecosystem functions and services in drying river networks (Sauquet et al., 2020; Acuña
et al., 2014; Leigh and Datry, 2017; Datry et al., 2017). An important reason is the spatial scale of both, observations and the
associated modelling results. Observations of drying patterns are often biased because of the installation of gauging stations in
larger rivers instead of smaller tributaries, which can result in misleading interpretations and an oversight of changing patters
in the entire river network (Yu et al., 2020). Döll et al. (2024) introduce a novel approach to estimating flow intermittence
across Europe. Despite increasing their spatial resolution to 15 arc-seconds ( 500 m), their global modeling approach cannot
achieve the high resolution needed to capture small intermittent reaches (Döll et al., 2024). Additionally, hydrological models
used for rainfall-runoff simulations at the catchment scale often generalize river networks, which can compromise fine-scale
details necessary to accurately represent reach-level dynamics. This limitation reflects the broader challenges of scaling in
basin hydrology when aiming on capturing the internal heterogeneity, especially at finer spatial scales (Blöschl and Sivapalan,
1995; Thompson et al., 2011). Another reason is linked to the projection of flow intermittence under climate change. There
are various notable studies assessing the impacts of climate change on hydrological processes at the river basin scale and
evaluating low flows under climate change Cammalleri et al. (2020); Krysanova and Hattermann (2017); Parey and Gailhard
(2022). However, limited data and models exist to quantify flow intermittence and drying patterns in a high spatio-temporal
scale.

To address these gaps, we present what is, to our knowledge, the first study assessing projections of flow intermittence
under various climate change scenarios, utilizing an ensemble of five global climate models across six European drying river
networks (DRNs). The DRNs are located in different biogeographic regions of Europe ranging from Southern Spain to Finland.
The spatio-temporal resolution allows to observe changes in flow intermittence at both scales, the river network (150 km$^2$ to



350 km$^2$) and the reach level (mean length ≈ 900 m). For this purpose, we use a hybrid modelling approach developed by
Mimeau et al. (2024), which combines physical-based hydrological modelling with machine learning techniques to predict
flow intermittence in European drying river networks. These models are forced with climate projection data (Devers et al.,
2023) to estimate and assess streamflow intermittence under climate change in the six pilot DRNs. Similarly to Mimeau
et al. (2024), in this study the term "flow intermittence" refers to two conditions, which are flowing phases and phases with
interrupted flow. The latter applies to both, a completely dry riverbed or disconnected pools.

In this paper we seek to use our results to answer the following questions:

- Are there changes in flow regimes, particularly in the transitions of reaches from perennial to intermittent? Where do
these changes occur—upstream, downstream, or randomly distributed—and under which emission scenarios might these
transitions take place, along with the timeframe for such changes?

- How are the spatio-temporal patterns of flow intermittence changing across the pilot-DRNs? This includes examining
both network-scale changes, such as the percentage of the river network experiencing intermittence, and reach-scale
changes, such as the number of days with dry conditions.

- What changes are occurring in the characteristics of drying events at the reach scale, focusing not only on the number of
events but also on the duration of drying events per year, as well as the timing of the first dry events?

- What are the potential evolutions of extreme drying events? This question involves analyzing the annual maximum extent
of ten consecutive dry days within the river network.

This work is embedded in the DRYvER project, aiming to integrate hydrology, ecology, and socio-economics to better
understand, predict and project changes in river flow regimes and their subsequent effects on biodiversity, ecosystem functions,
and services, with the goal of developing strategies and tools for mitigating the impacts of climate change on intermittent river
networks Datry et al. (2021). Within the scope of this collaboration, tools such as the DRYRivERS mobile application were
developed to monitor stages of flowing, disconnected pools and dry riverbeds (Truchy et al., 2023), which were used among
other crowdsourced data (Kampf et al., 2018) to calibrate the flow intermittence models developed (Mimeau et al., 2024).

## 2  Material and method

### 2.1  Study area

This study focuses on the six European drying river networks (DRNs) studied in the DRYvER project (Datry et al., 2021).
The studied DRNs are located in Croatia (HR), Czechia (CZ), Finland (FI), France (FR), Hungary (HU), and Spain (ES)
(Fig. 1). The sites are characterised by contrasting climates with a Mediterranean climate for the Genal (ES) and Butiznica
(HR) catchments, a pre-alpine climate for the Albarine catchment (FR), a continental climate for the Bükkösdi (HU) and
Velicka (CZ) catchments, and a boreal climate for the Lepsämänjoki catchment (FI) (Tab. 1).



These sites were monitored regularly during the DRYvER project by local DRN teams, with observations of the state of flow

in the river streams (Truchy et al., 2023), and several campaigns of biological samplings to monitor the impact of drying on

the metacommunity dynamics (Silverthorn et al., 2024; López-Rojo et al., 2024).

The six sites are characterised by regular drying up of part of the river networks, although the patterns and causes of drying

differ from one site to another. The Genal (ES) and Butiznica (HR) are the driest DRNs with a large proportion of river length

drying during summer months (June to September) due to meteorological and hydrological droughts. Besides, in the Butiznica

DRN Karst plays a major role in drying. The highly permeable substrate allows for substantial water storage in aquifers.

This permeability often includes larger openings, facilitating groundwater flow. After prolonged periods without precipitation,

groundwater levels can drop, leading to dry riverbeds or partial cessation of flow (where the river continues to flow, but partly

above and partly below the ground surface). For instance, in November 2021, the largest tributary to the Butiznica River, the

Radljevac stream, was flowing at the upper sampling location while dry at the lower part, indicating that the flow continued

underground, leaving the surface dry. Some sections of the main river in the Genal catchment experience drying due to water

abstraction for irrigation purposes. Flow intermittence in the Albarine DRN (FR) is caused by its geological characteristics

with seepage of the river streams into morraine deposit in the downstream part of the catchment and in karstic geology in the

upstream part of the catchments. Drying in the Velicka DRN is caused by a combination of human activities, such as water

abstraction and land use changes, and climatic factors related to its continental location, exacerbated by climate change with

drying events mostly occuring in late summer and autumn. Lepsämänjoki (FI) is the least intermittent DRN. Flow intermittence

is only observed in small headwater streams in July or August and drying events are shorter than in the other DRNs, covering

a few weeks at most.





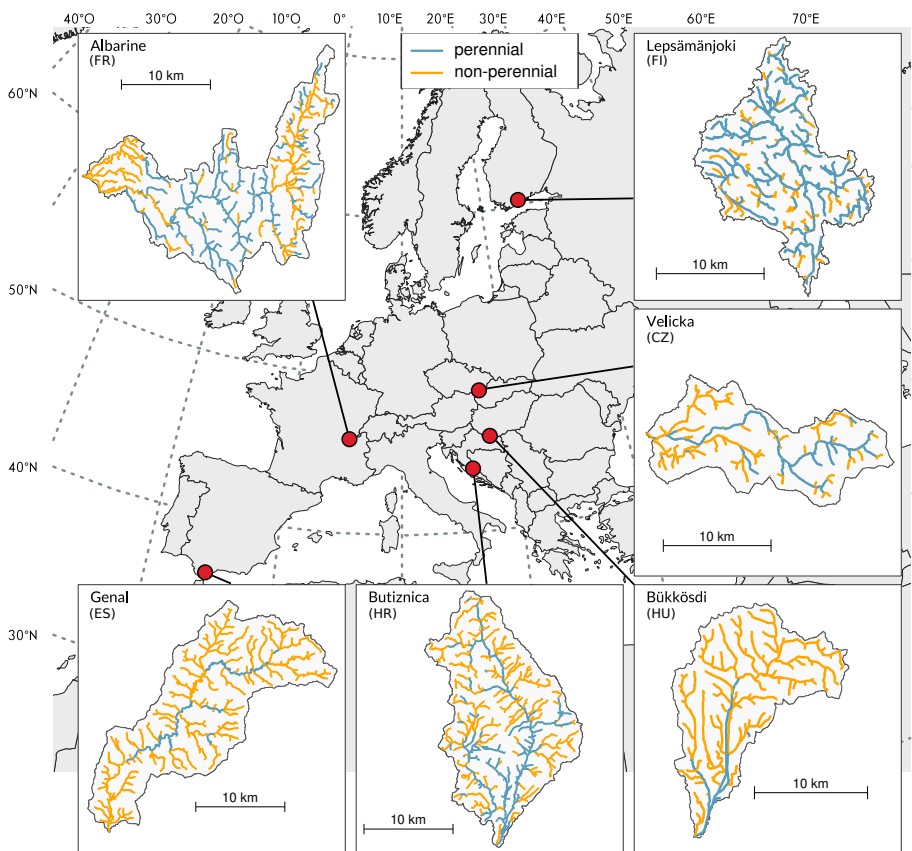

**Figure 1.** Flow intermittence regimes in the six studied DRNs. Flow intermittence regimes per reaches was obtained from the reconstruction simulation over the period 1985-2014 (see Section 2.3). A reach is considered as perennial if the mean of the annual number of dry days is below 1.

| Name | Country | Area | Outlet lat. | Outlet lon. | Range of elevations | Mean annual temperature | Mean annual precipitation |
|---|---|---|---|---|---|---|---|
| | | [km$^2$] | [°] | [°] | [m.a.s.l] | [°C] | [mm] |
| Albarine | France (FR) | 350 | 45.906 | 5.234 | 212 - 1497 | 10.0 | 1439 |
| Bükkösdi | Hungary (HU) | 190 | 45.989 | 17.928 | 106 - 500 | 11.6 | 750 |
| Butiznica | Croatia (HR) | 325 | 44.041 | 16.187 | 215 - 1563 | 9.8 | 1250 |
| Genal | Spain (ES) | 336 | 36.318 | -5.312 | 3 - 1718 | 15.9 | 743 |
| Lepsämänjoki | Finland (FI) | 208 | 60.238 | 24.984 | 8 - 145 | 5.6 | 899 |
| Velicka | Czechia (CZ) | 177 | 48.907 | 17.340 | 164 - 913 | 9.3 | 774 |

**Table 1.** Characteristics of the studied DRNs. Mean annual precipitation and temperature are computed from the ERA5-land reanalysis for the period 1991-2020.



## 2.2 Flow intermittence model

This section provides a summary description of the flow intermittence model used in this study, a more detailed description is
presented in Mimeau et al. (2024).

Flow intermittence is simulated using a hydrological model coupled to a random forest (RF) model (Fig. 2). The spatially
distributed JAMS-J2000 hydrological model (Kralisch and Krause, 2006) is used to simulate the streamflow as well as various
hydrological variables such as interception, actual evapotranspiration, water content in the soil and groundwater reservoirs,
etc. As the JAMS-J2000 model is unable to represent the drying up of rivers, it is coupled to a statistical RF model (Breiman,
2001) which predicts the daily state of flow using hydrological variables simulated by JAMS-J2000 (streamflow, groundwater
contribution to streamflow, evapotranspiration, saturation of the soil reservoir, saturation of the groundwater reservoir) as well
as hydro-meteorological variables (precipitation, temperature) and reach characteristics (drainage area, slope, landuse, type of
soil, type of geology) as predictors. The RF model is trained using state of flow observations collected in the DRNs and then
used to predict the state of flow in every reach of the DRNs according to the meteorological and hydrological conditions of the
previous days.

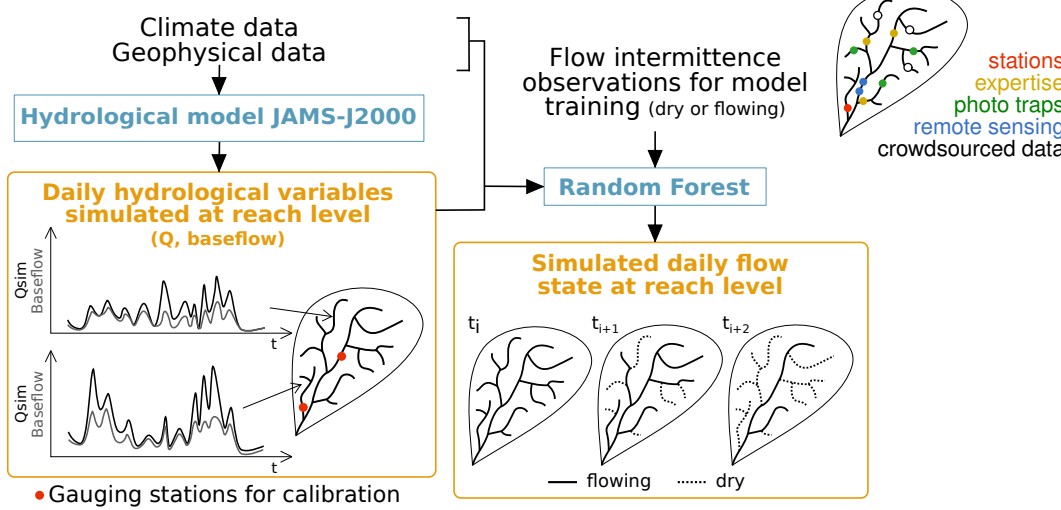

**Figure 2.** Principles of the flow intermittence model (adapted from Mimeau et al. (2024))

The flow intermittence model was applied in the six studied catchments using, as much as possible, the same datasets and
methodology. The JAMS-J2000 model was calibrated and validated in the DRNs on the period 1995-2020 using gauging
stations data and the RF model was trained separately in the six DRNs during the period 2005-10-01 to 2022-04-30 using
multiple datasets of observed state of flow during the training period.

All information regarding the set-up of the JAMS-J2000 hydrological model (spatial data, parameter values, calibration
method, and model performance at the gauging stations) are available in Mimeau et al. (2024) for the Albarine (FR), Genal
(ES), and Lepsämänjoki (FI) DRNs, and in Künne et al. (2022) for the Bükkösdi (HU), Butiznica (HR), and Velicka (CZ)





DRNs. Details on the observed state of flow data used to train the RF models are presented in supplementary material in Section S1.

In order to evaluate the flow intermittence model performance, the RF model was trained with 75 % of available observed data selected randomly over the period 2005-2021, and was evaluated on the remaining 25 %. Fig 3 shows the ranges of the probability of detection of drying events (POD) and false alarm ratio (FAR) for an ensemble of 20 training and evaluation runs of the RF model. The POD and FAR criteria are computed as follow:

$$POD = \frac{a}{a + c}$$


$$FAR = \frac{b}{a + b}$$

with $a$ the number of dry observations correctly simulated by the model, $b$ the number of flowing observations that were simulated as dry, and $c$ the number of dry observations that were simulated as flowing. Fig 3 shows good performances of the flow intermittence model for the Albarine (FR), Bükkösdi (HU), Lepsämänjoki (FI), and Velicka (CZ) DRNs, with POD above
90 % and FAR around 5 %. For the Butiznica (HR) and Genal (ES) DRNs, the average POD is respectively equal to 71 and 65 % and average FAR to 12 and 19 %. The poorer performance of these two DRNs can be explained by the lack of observed data to train the RF model.

To reduce the number of simulations for the projection scenarios and to use the most of the observed data, only one member of the RF models trained with 100 % of the observed data is used in the rest of the study. Fig S2 shows the simulated seasonal
drying pattern with the selected member for each DRNs compared to the seasonal drying patterns obtained with the full RF models ensemble.

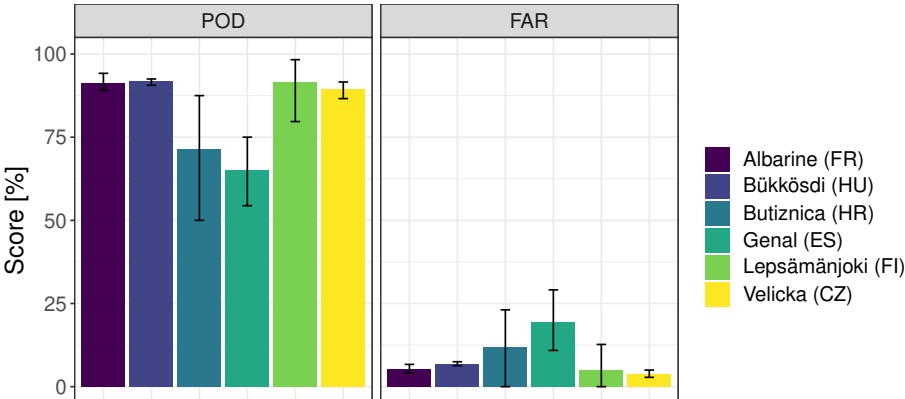

**Figure 3.** Performance of the flow intermittence model on the training period (2005-2021) for the 6 DRNs. Bars show the average value for the 20-member ensemble and error bar the minimum and maximum values for the 20-member ensemble. POD: probability of detecting a drying event, FAR: false alarm ratio.



## 2.3 Reconstruction simulations

To simulate flow intermittence during the past-observed period (1960-2021), the flow intermittence model was forced with the ERA5-Land reanalysis (Muñoz-Sabater et al., 2021). The following variables were extracted at an hourly time step and

0.1° spatial resolution over the six studied catchments between 1950-01-01 and 2021-12-31, and then aggregated at daily time step:

- 2m air temperature (°C),

- 2m dew point temperature (°C),

- 2m relative humidity (%),

- 10m u and v wind speed components (m/s),

- surface pressure (Pa),

- surface solar radiation downwards (W/m$^2$),

- surface thermal radiation downwards (W/m$^2$),

- total precipitation (mm).

The daily variables are then used to calculate the reference evapotranspiration using the Penman-Monteith equation (Allen et al., 1998).

    The model is then forced with the daily time series of precipitation, temperature and reference evapotranspiration. Simulations are run over the period 1950-2021, and results analyzed over the period 1960-2021 removing the first 10 years of initialization.

## 2.4 Future projections

Climate projections from the Inter-Sectoral Impact Model Intercomparison Project (ISIMIP3a and ISIMIP3b, http://www.isimip.org, last access: 19 August 2024) were used. This study uses projections from five CMIP6 General Circulation Models (GCMs) (GFDL-ESM4 / IPSL-CM6A-LR / MPI-ESM1-2-HR / MRI-ESM2-0 / UKESM1-0-LL) selected in the ISIMIP project on the basis of their historical performance and to reflect the climate sensitivity of the full CMIP6 ensemble (Frieler et al., 2024) .

Three SSP scenarios are considered: SSP1-2.6 Sustainability, SSP3-7.0 Regional rivalry, and SSP5-8.5 Fossil-fuelled development.

    The data were extracted at a daily time step between 1971 and 2100. The variables extracted are the same as for the ERA5-land re-analysis and evapotranspiration was calculated with the Penman-Monteith equation. The GCMs projections, with a 0.5° spatial resolution, were downscaled to obtain high resolution projections with a 0.1° spatial resolution using the analogy

method which is fully described in Devers et al. (2023). The downscaling method produced 20-members ensembles for each





combination of GCMs and SSP scenarios for the 6 studied catchments. The dispersion of the 20 members relates to the uncertainty in the analogue dowsncaling method. As stated by Devers et al. (2023, see e.g. their Fig. 4), this uncertainty is rather negligible with respect to both the SSP uncertainty and the GCM uncertainty. As a consequence, and in order to limit the computation time, only one member is considered in the following. Moreover, the 20-member ensemble had been built to be unbiased as a whole, with individual members being either slightly too wet or slightly too dry systematically over time. The $10^{th}$ member at the centre of the distribution was therefore chosen to minimize the bias in downscaled projections.

The high-resolution projections of daily temperature, precipitation and evapotranspiration produced by the downscaling method were used to force the flow intermittence model over the 1971-2100 period, including a 14 years period of initialization.

To clarify the terminology used throughout the paper, we provide the following definitions:

– Reconstruction period: Refers to simulations of flow intermittence during the past-observed period (1960-2021) using climate reanalysis data from ERA5-land as forcing data. These simulations aim to reconstruct historical hydrological conditions based on observed climate data.

– Historical period (1985 - 2014): Refers to simulations of flow intermittence under historical climate conditions, utilizing climate projections from the ISIMIP phases 3a. These simulations are based on climate data obtained from the 5 climate models used and serve as a baseline for comparison with future scenarios.

– Future projections/SSP Scenarios (2015 - 2100): Refers to different Shared Socioeconomic Pathway (SSP) scenarios used to represent plausible future socioeconomic and climate trajectories. In this study, we consider three SSP scenarios: SSP1-2.6 (Sustainability), SSP3-7.0 (Regional rivalry), and SSP5-8.5 (Fossil-fuelled development), each representing different assumptions about future societal and economic development pathways and corresponding GHG emissions. These simulations are conducted for each GCM across the three SSP scenarios, resulting in a total of 15 simulations (5 GCMs × 3 SSP scenarios). In the Results section, the flow intermittence projections are analysed for two time horizons: the medium term (2041-2070) and the long term (2071-2100).

## 2.5 Flow intermittence indicators

To better quantify and evaluate the modelled outputs of flow intermittence at reach-scale, a suite of flow intermittence indicators has been developed. This was a joint effort within the DRYvER project consortium to ensure that these indicators provide insights into the impact of flow intermittence on biological, chemical and socio-economic processes and potential change signals under present and projected climate conditions (Künne and Kralisch, 2021). In this study, a subset of these indicators were used to assess hydrological patterns and projected changes. Their definitions are listed in Table 2. They are either quantifying information in relation to the reach-scale, i.e. ID 1 - 6 or to the river network-scale, i.e. ID 7 - 11. At the reach scale, flow intermittence is quantified differently by several indicators on a monthly and annual basis. The number of days with dry ($ConD$) and flowing ($ConF$) conditions (ID1 in Table 2) allows to track how frequently a river reach experiences these states over different time scales, which helps identify seasonal patterns and long-term trends. In addition to the frequency, the duration of dry ($DurD$) and flowing ($DurF$) conditions (ID2 in Table 2) measures the number of consecutive days a reach remains in these





states within a year. This provides insights into the persistence of specific hydrological conditions. The absolute number of
drying ($numFreDr$) and rewetting ($numFreRW$) events (ID3 in Table 2), along with their relative frequencies ($percFreDr$
and $percFreRW$; ID4 in Table 2), help to understand the dynamics of flow intermittence by counting transitions between
dry and wet states. On an annual basis, we also examine the day of the first drying event ($FstDrE$; ID5 in Table 2), which
records the Julian day when the first drying event occurs each year. This is crucial for understanding changes in the onset of dry
conditions in reaches. Furthermore, the intermittence regime ($IntReg$; ID6 in Table 2) classifies the flow regime of a reaches
as either perennial or intermittent based on the average number of dry days per year: This threshold can be adapted for each
DRN, based on the different climatological and environmental characteristics of the basin. In this study, we define a reach as
perennial if it experiences, on average, less than one day of flow intermittence over a 30-year period. Conversely, a reach is
considered non-perennial if it experiences more than one day of flow intermittence on average over the same period.

At the network scale, several indicators help to assess spatial patterns of drying and connectivity. The proportion of network
length with flowing conditions ($RelFlow$; ID7 in Table 2) measures the percentage of the river network experiencing flowing
conditions. Similarly, the proportion of network length with flow intermittence ($RelInt$; ID8 in Table 2) quantifies the extent
of intermittent flow conditions within the network. Besides, annually and at the network-scale the average longitudinal length
of dry ($LengthD$) and flowing ($LengthF$) reaches (ID9 in Table 2) is calculated, providing a spatial dimension to flow inter-
mittence by determining the average length of reaches experiencing permanent dry or flowing conditions each year. In addition,
the patchiness of steady and intermittent flow conditions ($PatchC$; ID10 in Table 2) assesses the proportion of the river net-
work where flow conditions change between adjacent downstream reaches computing the variability of flow conditions. The
indicator can help to identify where flow conditions change frequently along the stream, which can be crucial for understanding
habitat connectivity and ecological dynamics. Finally, the extreme drying events ($RelInt10max$; ID11 in Table 2) describe
the annual maximum fraction of the river network that dries up over a 10-day period, helping to identify and quantify extreme
drying events that can have significant impacts on aquatic ecosystems and water quality.

These indicators allow a comprehensive assessment of flow intermittence across different temporal and spatial scales, pro-
viding valuable insights into the dynamics of drying river networks under current and future climate conditions. The calculation
of these indicators is explained more detailed within the supplementary material S 3.

## 3 Results

### 3.1 Climate and hydrological projections

Figure 4 shows the evolution of climate and hydrological variables in the DRNs for the three SSP scenarios. Projections of the
mean annual temperature and precipitation are directly obtained from the downscaled climate projections, and projections of
mean annual actual evapotranspiration and streamflow are simulated with the JAMS-J2000 models. Flow trends are given for
river sections corresponding to the location of gauging stations (see Fig. S3), where streamflow is known to be perennial. The
mean annual temperature is increasing in the DRNs and across the 3 SSP scenarios. For the SSP1-2.6 scenario, temperatures
rise during the first half of the 21$^{st}$ century, then start to fall, with an increase in 2071-2100 compared to 1985-2014 ranging





**Table 2.** Indicator suite and their definition at the reach scale and network scale

| ID | Indicator name | Abbreviation | Definition |
|---|---|---|---|
| | | Reach scale | |
| 1 | Number of days with dry/flowing conditions | $ConD$/$ConF$ | Number of days with dry and flowing conditions per month [d/m] and year [d/a] |
| 2 | Duration of dry/flowing conditions | $DurD$/$DurF$ | Number of consecutive days with dry/flowing conditions per year [d/a] |
| 3 | Absolute drying/rewetting events | $numFreDr$/ $numFreRW$ | Absolute number of drying/rewetting events [n] |
| 4 | Relative drying/rewetting frequency | $percFreDr$/ $percFreRW$ | Relative frequency of drying/rewetting events [%] |
| 5 | Day of first drying event | $FstDrE$ | Julian day of the first drying event per year [1 - 366] |
| 6 | Intermittence regime [perennial or intermittent] | $IntReg$ | Threshold for the average number of dry days per year ($ConD$) to characterise a permanent or intermittent flow regime. If $ConD$ < threshold, then the regime is perrenial, otherwise the regime is intermittent |
| | | Network scale | |
| 7 | Proportion of network length with flowing conditions | $RelFlow$ | Proportion of model-derived river length with flowing conditions [%] |
| 8 | Proportion of network length with flow intermittence | $RelInt$ | Proportion of model-derived river length with flow intermittence [%] |
| 9 | Annual average length of dry/flowing reaches | $LengthD$/$LengthF$ | Average longitudinal model-derived river length with permanent dry/flowing conditions [m/a] |
| 10 | Patchiness of steady and intermittent flow conditions | $PatchC$ | Proportion of model-derived reach length with changing flowing and intermittent conditions compared to adjacent downstream reaches [%] |
| 11 | Extreme drying event | $relIint10max$ | Represents the annual maximum fraction of river network dried up over a 10-days period [%] |





between +1.2 and +2.0 °C. For the SSP3-7.0 and SSP5-8.5, temperatures keep rising until 2100. The temperature increase by 2071-2100 is ranging between 3.2 and 4.6 °C for SSP3-7.0 and between 4.1 and 5.6 °C for SSP5-8.5.

The trends of precipitation vary depending on the scenario and DRN. For scenario SSP1-2.6, there is no clear trend in the 6 DRNs. For the SSP3-7.0 and SSP5-8.5 scenarios, there is a north-south gradient in precipitation changes, with a slight increase in cumulative annual precipitation in Lepsämänjoki (FI) (+4 % for SSP3-7.0 and SSP5-8.5), a slight decrease in the Albarine (FR) and Velicka (CZ) basins (between -5 and -9 % for SSP5-8. 5), a more marked downward trend for Bükkösdi (HU) and Butiznica (HR) (-7 % for SSP3-7.0 and -14 % for SSP5-8.5), and finally a sharp decrease in the Genal basin (ES) (-26 % for SSP3-7.0 and -33 % for SSP5-8.5).

Due to the temperature increase, evapotranspiration tends to increase for the 3 SSP scenarios for Lepsämänjoki (FI), Velicka (CZ), Albarine (FR), and Butiznica (HR). The increase in evapotranspiration is greatest for the Lepsämänjoki basin (FI) (+17 % for SSP5-8.5 by 2071-2100), due to the combined effect of rising temperatures and precipitation. For the Bükkösdi (HU) and Genal (ES) basins, evapotranspiration decreases by 2071-2100 for the SSP3-7.0 (-3 and -12 %) and SSP5-8.5 (-9 and -17 %) scenarios because the evaporative demand can no longer be met as a result of falling precipitation.

Streamflow is decreasing sharply for all DRNs and for the 3 SSP scenarios. The Lepsämänjoki basin (FI) dries out the least, with a decrease in streamflow ranging between -10 and -17 % depending on the SSP scenarios by the end of the 21[st] century as the increase of evapotranspiration is partly compensated by the increase in precipitation. For the other DRNs, the decrease in streamflow at the end of the century is particularly significant for the SSP3-7.0 and SSP5-8.5 scenarios. The mean annual streamflow in the Bükkösdi (HU), Genal (ES), and Velicka (CZ) catchments could be reduced by more than half by 2071-2100

for SSP5-8.5. All the projection values for temperature, precipitation, evapotranspiration and streamflow to 2071-2100 for the 3 SSP scenarios and 6 DRNs are given in Table S2.





**Figure 4.** Projections of annual temperature (T), precipitation (P), actual evapotranspiration (ET), and streamflow at the main gauging station (QA) (see Fig. S3 for the locations of the gauging stations) for the 3 SSP scenarios (in anomaly compared to the historical reconstruction). Thin lines represent the results obtained for the 5 downscaled GCM models, and wider lines the smoothed average of the 5 GCMs.

## 3.2 Flow regime transition

In this section we analyze the evolution of flow regimes in river systems under climate change scenarios. As a reminder, in this study a reach is defined as perennial if it has less than one day of dryness on average per 30-year period, and non-perennial in

the opposite case. Figure 5a shows the majority flow regime simulated with the SSP5-8.5 scenario for the period 2071-2100 in





the DRNs for the 5 GCMs, and shows in particular the reachs for which there is a transition from a perennial to a non-perennial regime compared with the historical period, and Figure 5b show the evolution in time of the proportion of river length with a perennial flow regime for the 3 SSP scenarios. For the 6 DRNs, the results show a decrease in the projected length of rivers with a perennial flow regime in 2041-2070 and 2071-2100 compared to 1985-2014.

In the Genal (ES) and Velicka (CZ) DRNs, a very high proportion of the river networks could transition from a perennial to a non-perennial regime by the end of the 21$^{st}$ century. The results show a decrease of -57 % for SSP3-7.0 and -60 % for SSP5-8.5 of length of perennial rivers in Velicka and a decrease of -77 % for SSP3-7.0 and -93 % for SSP5-8.5 in Genal. For the Velicka DRN, the flow regime transitions occur mainly on the downstream part of the catchment, notably on sections of the main river. For the Genal DRN, transitions occur along the main river, making almost all the river network river intermittent.

Lepsämänjoki (FI) and Butiznica (HR) also have a significant proportion of their river network switching to a non-perennial regime by 2071-2100 (in average -19 % of perennial river length for Lepsämänjoki and -39 % for Butiznica with SSP5-8.5), but with different spatial patterns of regime shift. For the Lepsämänjoki DRN the flow regime transitions occur mainly on the headwater reach, whereas in Butiznica the transitions occur for all types of reaches, from the headwaters to the main river sections.

There is less projected transitions from perennial to non-perennial regimes in the Albarine (FR) and Bükkösdi DRNs. For the Bükkösdi DRN, this is mainly due to the fact that the river network is already highly intermittent during the historical period. For the Albarine DRN, a few reaches mainly located in the two main areas of flow intermittence (downstream and all the way upstream) show a switch in their flow regime with SSP5-8.5. The low proportion of river section with a regime transition in the Albarine compared to similar DRNs such as Velicka (CZ) and Butiznica (HR) could be due to the flow intermittence model

structure (see section 4.5).





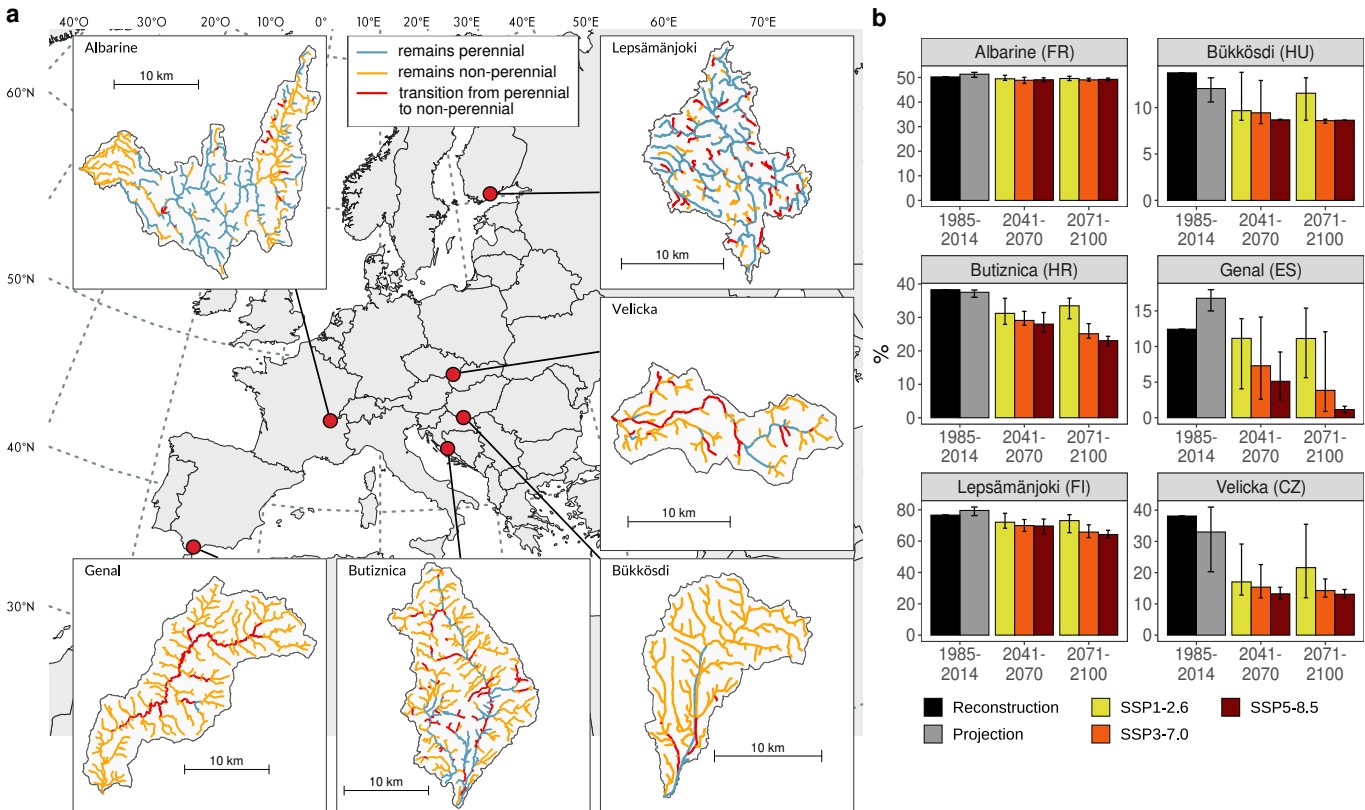

**Figure 5.** a) The evolution of flow regimes (2071-2100 vs 1985-2014) for SSP5-8.5 (color red shows the reaches for which 3 or more GCMs shows a transition from a perennial to a non-perennial regime). b) Evolution of the proportion of river length with a perennial flow regime for scenarios SSP1-2.6, SSP3-7.0, SSP5-8.5 (bar plots show the average of the 5 GCMs and error bars the range of the 5 GCMs).

## 3.3 Evolution of flow intermittence in the river networks

As a result of the decrease in streamflow simulated by the JAMS-J2000 hydrological model, the flow intermittence model simulates a strong increase in the average annual length of dry rivers by 2100 (Fig. 6). For the SSP1-2.6 scenario, the annual length of dry river network keeps increasing until 2050 and then stabilizes or slightly decrease in the second half an the century. For the scenarios SSP3-7.0 and SSP5-8.5, drying continues to increase until 2100.

The intensification of drying up is greatest in the Bükkösdi (HU) DRN, with 94 km (+43 %) more rivers drying up on average per year by 2071-2100 than in the historical period for SSP5-8.5. Lepsämänjoki (FI) is the DRN with the smallest increase of drying with +8 km (+3 %) of dry rivers in average by 2071-2100. For the other DRNs, the anomaly of annual dry river length ranges between +6 and +13 % for the SSP5-8.5 scenario: +46 km (+13 %) for Genal (ES), +13 km (+8.5 %) for Velicka (CZ), +35 km (+8.4 %) for Butiznica (HR), and +24 km (+6.1 %) for Albarine (FR).

Figure 7 shows the evolution of the seasonal patterns of drying in the DRNs by 2071-2100. For the Lepsämänjoki (FI) DRN, drying become more intense during the summer season. July stays the driest month in the projections, but the drying





up season is longer starting in May and ending in June versus June to August for the historical period. For the Albarine (FR) DRN, the drying period is also increased, with it's maximum of drying up shifted to the end of the summer and the end of

drying up later, until October. For the Butiznica (HR) DRN, the projections show a strong modification of the seasonal pattern of drying with intensification of drying during the second half of the year. The driest period is shifting from May to September for the historical period to June to November for the period 2071-2100. For the Velicka (CZ) and Bükkösdi (HU) DRNs, the intensification of drying is fairly homogeneous along the year. For these two DRNs, projections show an increase in drying both during the summer and winter seasons. The Genal (ES) DRN, is the only DRN showing a a higher intensification of drying

during the winter season than during the summer season. This is due to the fact that the Genal river network is already very dry in summer in the historical season (around 70 % of the river network dries up in July-August) and cannot dry up much further in the future, even for the SSP5-8.5 scenario.



**Figure 6.** Anomaly of the mean annual length of dry river network (in km) compared to the historical reconstruction for the 3 SSP scenarios. Thin lines represent the results the 5 downscaled GCM models, and wider lines the smoothed average of the 5 GCMs. The y-axis scale is different for each DRN



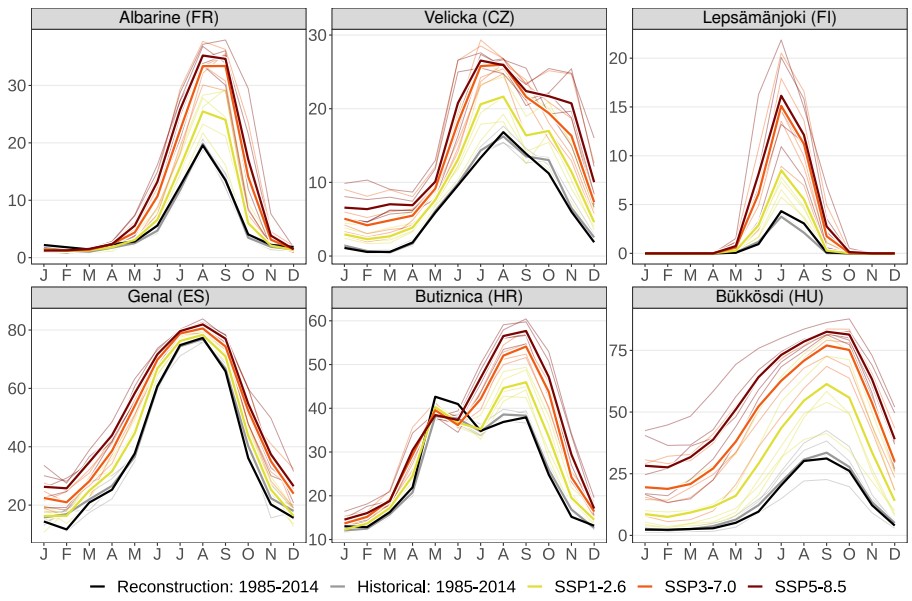

**Figure 7.** Evolution of the seasonal pattern of proportion of dry river length (in %) by 2071-2100 for 3 SSP scenarios. Thin lines represent the results for the 5 downscaled GCM models, and wider lines the average of the 5 GCMs.

## 3.4 Characteristics of dry spells at the reach scale

This section focuses on the evolution of drying locally at the reach scales. Here a dry spell is defined as consecutive days with a dry condition in a reach without taking into account the flowing condition in the other reaches during the same period. To do this, we consider 4 classes of reaches based on the average mean annual number of dry days ($ConD$) from the reconstruction simulation (1985-2014) (Fig. 8b): class 1 for reaches with $ConD$ < 5 days/yr, class 2 with $ConD$ between 5 and 30 days/yr, class 3 with $ConD$ between 30 and 120 days/yr, and class 4 with $ConD$ > 120 days/yr.

Figure 8a shows the anomaly in the first annual day of drying ($FstDrE$), the annual number of days with dry spells ($numFreDr$), and in the maximum duration of dry spells ($DurD$) of the projection simulations compared to the reconstruction simulation. Firstly, the projection simulations show that $FstDrE$ tend to decrease for the 3 SSP scenarios at the horizon 2071-2100 compared to the historical period 1985-2014, which means that there is a general trend in the DRNs towards earlier dry conditions in the future with an increasing intensity according to the severity of the SSP scenario. However, there is no general pattern that emerges according to the reach classes in all the DRNs. For the Velicka basin (CZ), the anticipation of dry spells becomes greater at the end of the centuries for the driest reach classes. For the Lepsämänjoki basin (FI), there is little difference between the 3 reach classes. For Butiznica (HR), it is classes 1 and 2 that show the greatest changes towards earlier dry spells, whereas it is classes 2 and 3 for the Albarine (FR) and Bükkösdi (HU). And for Genal (ES), the anticipation of dry periods is more marked for class 3, but the results also show a trend towards later dry periods for class 1.





Regarding the evolution of $DurD$, Fig. 8a shows a massive increase in the 6 DRNs, correlated with the reach class: the more

the reach dries up in the historical period, the greater the increase in the maximum duration of drying in future projections. For the reach classes 3 and 4, $DurD$ barely exceeds a few weeks for the projections simulations in the historical periods, but regularly becomes greater than 1 month in the DRNs, and even several months for the Bükkösdi (HU), Butiznica (HR), and Velicka (CZ) DRNs, for the SSP3-7.0 and SSP5-8.5 scenarios by 2071-2100.

In contrast, the number of annual dry spells changes little with the climate change scenarios. For the reach classes 1 and

2, $numFreDr$ tends to slightly increase in the projections by 2070-2100 compared to the historical period. Trends are more contrasted for class 3, and for class 4 there is no change for the Butiznica (HR) and Velicka (CZ) and a decreasing trend for Albarine (FR), Bükkösdi (HU), and Genal (ES) DRNs.

Overall, the projections show two main types of change in intermittence in the DRNs at the horizon 2071-2100. Reaches characterised by infrequent or moderate drying over the historical period (classes 1 and 2) will tend to become more intermit-

tent, with slightly more frequent and longer drying events. In the case of reaches that dry up frequently and for a long time in the historical period (classes 3 and 4), the duration of drying up will tend to be much longer, reducing the alternation between phases of flowing and dry conditions and leading to prolonged periods of continuous dry spells. In both cases, dry spells will tend to start earlier in the year, with dates of first dry condition shifting to earlier in the year by a few weeks to more than a month, depending on the DRNs and SSP scenarios.





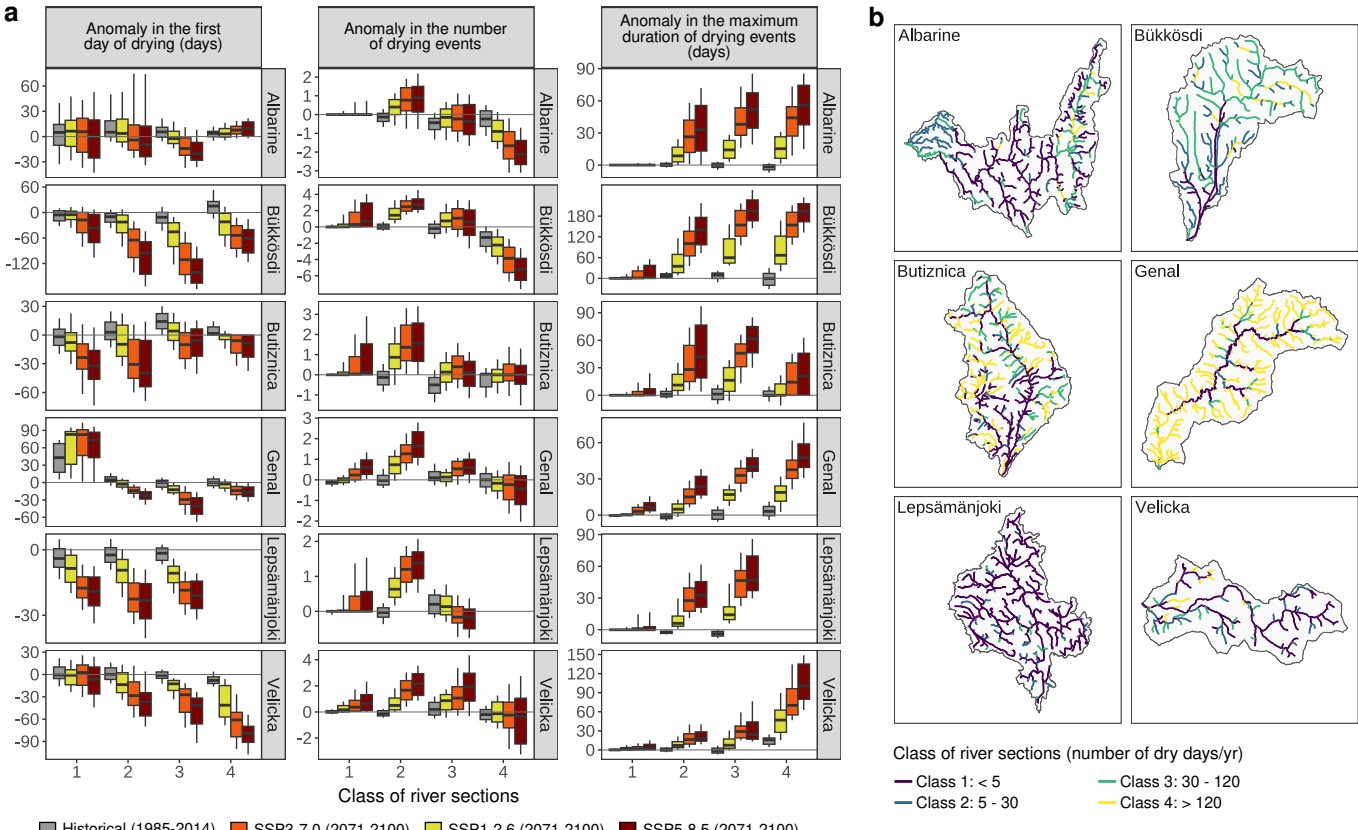

**Figure 8.** Evolution of the $FstDrE$, $numFreDr$, and $DurD$ indicators (in anomaly compared to the historical reconstruction (ERA5-land 1985-2014) per reach class (1 to 4). Reach classes are determined from the yearly $ConD$ indicator averaged over the period 1985-2014 for the reconstruction simulation (ERA5-land). Boxplots whiskers range from the 10[th] to the 90[th] percentiles.

## 3.5 Extreme droughts

This section focuses on extreme drying or droughts, i.e. periods with intense and widespread drying across the entire river networks. To do this, we analyze the evolution of the relinInt10max indicator, which represents the annual maximum fraction of river network dried up over a 10-days period.

Figure 9a shows the dates and spatial extent of the maximum drought event for the reconstruction simulation over the period 1980-2022 for each DRN. In the 6 DRNs, the most intense droughts happened during the last 10 years of the 1980-2022 period. For the Genal (ES) and Bükkösdi (HU) DRNs, more than 80 % of the river network is simulated as dry during the maximum drought event. For Velicka (CZ), around two-thirds of the river network dried up during the maximum event, around half for Albarine (FR) and Butiznica (HR), and a quarter for Lepsämänjoki (FI).

Figure 9b shows the projections of the annual maximum drought event ($RelInt$10max) for the 3 SSP scenarios. For the 6 DRNs, all the projection simulations show an increase in the intensity of extreme drought both in the long and medium term.





Projections show that the historical drought maxima presented in Figure 9a will be exceeded on a regular basis by the end of the century for the SSP3-7.0 and SSP5-8.5 scenarios for Bükkösdi (HU), Butiznica (HR) and Lepsämänjoki (FI).

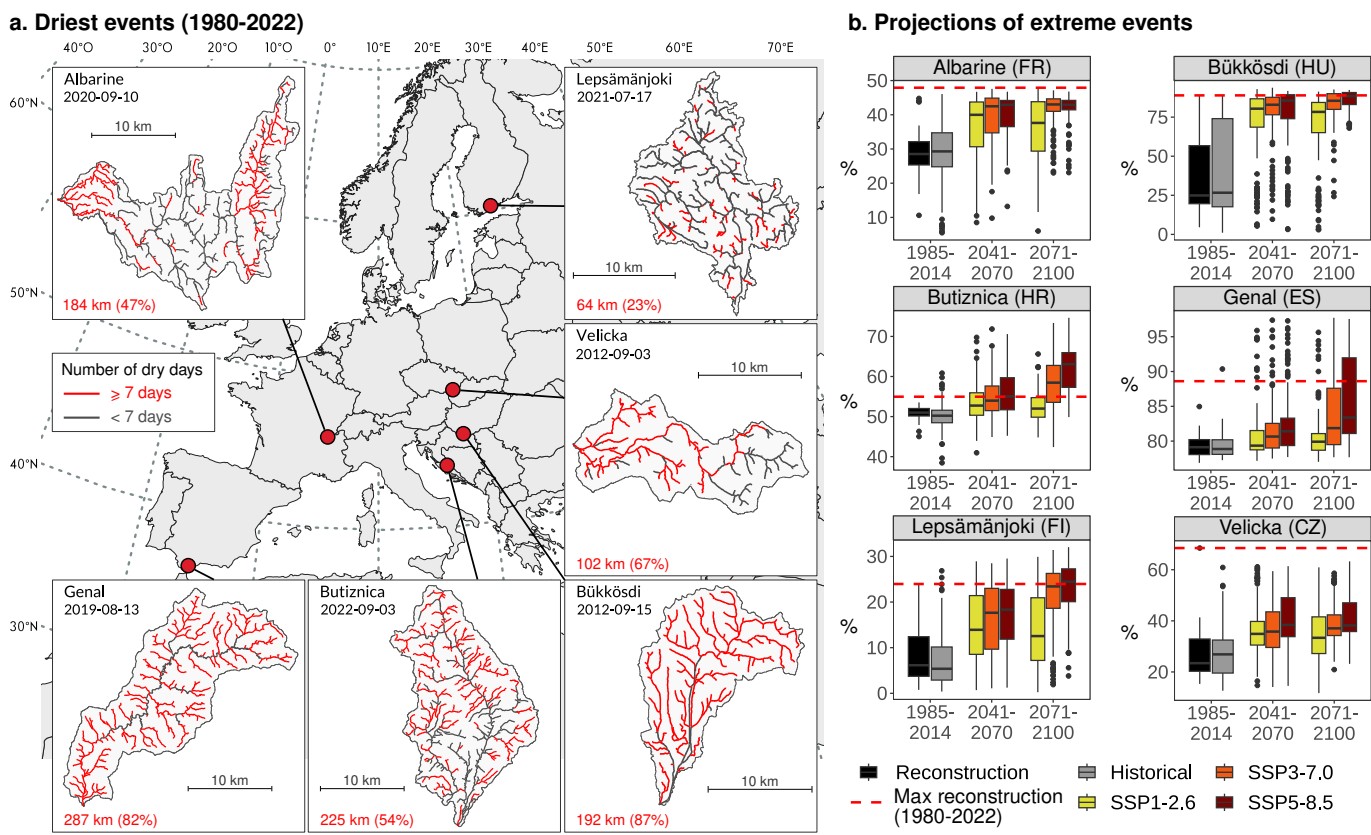

**Figure 9.** a. State of flow in the DRNs during the driest 10-days event of the reconstruction simulation (1980-2022 with ERA5land), red: reaches that dried during more than 7 days during the event. The starting date of the event as well as the length of river network with a drying above 7 days is indicated at the bottom of each panel. b. Evolution of $RelInt10max$ for the 3 SSP scenarios compared to the historical period. The horizontal red dashed line shows the $RelInt10max$ value corresponding to the historical driest event.

## 4 Discussion

### 4.1 Evaluation of the reconstruction simulations

This study is one of the first attempting to simulate the long term evolution of drying at the level of river reaches. One main difficulty is to assess the ability of a flow intermittence model to predict realistic spatio-temporal patterns of drying in river systems. Firstly because observed data of flow intermittence are point mesurements in both, space (time series from a few sites that are insufficient to represent the entire network) and time (while participatory science or remote sensing can provide well-distributed spatial observations, those are typically infrequent and discontinuous over time). In addition, measures to





observe the intermittence of watercourses have only recently been put in place, most commonly after 2010. As a result, the available the available time series are not long enough to observe long-term changes in the state of watercourse flow. Of the 6 study sites, only the data from the Saint-Denis-en-Bugey gauging station located in the Albarine basin (FR) has been recording observations on an intermittent section of river for several decades (since 1957). As a result, there is almost no quantitative observed data to characterise past spatial and temporal changes in the state of flow in river systems and evaluate the modelling.

390       This study was conducted in collaboration with teams of local researchers specialising in freshwater ecology in drying rivers systems. Through their field sampling activities, these teams have acquired in-depth knowledge of the intermittent characteristics of these basins. Thanks to their expertise, local researchers were able to provide maps of the flow regime in the Albarine (FR), Bükkösdi (HU), Genal (ES), and Velicka (CZ) river networks (Fig. S4). These data are subjective, since they are based on the researchers' perceptions, but they provide useful information for assessing the intermittence model's ability to represent

the drying-up dynamics observed in the field. Comparison between Fig. S4 and Fig. 1 shows that the model seems to provide a good representation of the intermittency regimes in the Albarine (FR) and Velicka (CZ) DRNs for the reconstruction period. For the Genal DRN (ES), the observation data is spatially very fragmented and relates only to recent years (2020-2023), so it is more difficult to assess the model on the basis of this data. However, the observations seem to $ConF$irm the simulated pattern, with a main perennial river and intermittent tributaries. The section of the main river simulated as intermittent on Fig. 1

in the Genal DRN comes from the value of the threshold set to define the perennial or intermittent regimes (taken here at 1 dry day/yr on average over a 30-year period), by taking a threshold equal to 2 dry days/yr this section of river is simulated as perennial (the uncertainty related to this threshold is further discussed in Section 4.2). For the Bükkösdi DRN (HU), flow intermittence is clearly overestimated by the model which simulates almost all tributaries to the Bükkösdi river as intermittent, whereas local expertise show that there are more river sections with a perennial flow regime. This error possibly comes from a

misclassification of flowing and dry events in the streamflow timeseries (see location of the gauging stations in Fig. S3). A first reason is the use of a common threshold for the 6 DRNs of 0.005 m³/s to identify dry days in the observed streamflow time series, which led to overestimate the number of observed dry events in the Bükkösdi DRN. The second reason comes from zero-flows that are possibly erroneous in the measured streamflow timeseries (Zimmer et al., 2020).

## 4.2   Definition of perennial and non-perennial regimes

There is no consensus among studies on a threshold for defining perennial and non-perennial flow regimes. Snelder et al. (2013) considers as perennials rivers with zero days of drying over the entire period, Messager et al. (2021) uses a threshold of one day per year on average, while other studies use thresholds of five days per year on average (Costigan et al., 2017), or up to 18 days per year over on average (Van Meerveld et al., 2020).

        Threshold values ranging from zero to 20 days per year were tested to analyze the sensitivity of this study's results to the

definition of non-perennial rivers (Fig. 10). Fig. 10a shows that the proportion of perennial rivers during the historical period can be very sensitive to the threshold value which ranges from 16 to 58 % for the Velicka DRN (CZ) and 71 to 90 % for the Lepsämänjoki DRN (FI) when the threshold value ranges between 0 and 5 days/yr on average. Sensitivity to the threshold value varies between DRNs but for all DRNs there is a significant difference between considering a threshold equal to zero





or equal to one dry drays per year. In average for the 6 DRNs, 7.6 % more river length is considered as perennial with a
threshold of one day per year comparing to a threshold of no dry day at all. The threshold value also impacts the projected
changes in the proportion of perennial rivers under climate change scenarios (Fig. 10b). The projections of the river lengths
switching from a perennial to a non-perennial regime at the horizon 2071-2100 for the SSP5-8.5 scenario are very sensitive to
the threshold value. For example, for the Velicka DRN, depending on whether a threshold of 0 or 5 days per year is considered,
the percentage of the river network switching to a non-perennial regime by the end of the century varies from 8 to 32 %.

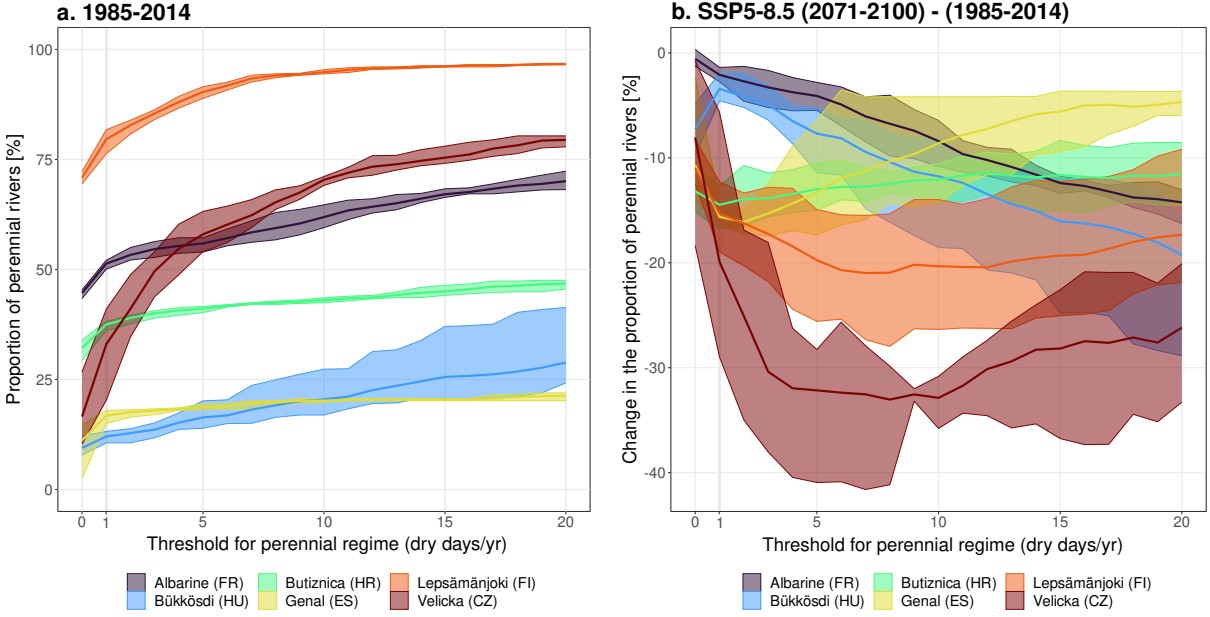

**Figure 10.** a. Proportion of perennial rivers for the 1985-2014 period, b. Change in the proportion of perennial rivers in 2071-2100 compared
to 1985-2014 according to the threshold to the threshold for defining a perennial regime (average, minimum and maximum values of the 5
GCMs).

## 4.3  Conceptualisation of extreme events

The concept of what constitutes an 'extreme' event varies based on the historical flow regime of the river reach. For example, a
single dry day could be considered extreme for a historically perennial reach, while two months of dryness might be moderate
for a reach that typically dries 50% of the time. By coupling temporal and spatial drying patterns, we aim to identify possible
extreme events at the network scale, such as those observed in the Albarine River in 2022 (Sarremejane et al., 2022; Datry
et al., 2023).

In Czechia, dry episodes shorter than seven days are considered mild for near-perennial streams, with limited impact on
biota, as refugia are not destroyed, allowing for quick recovery. This threshold has proven effective, achieving about 90 %
accuracy in distinguishing between intermittent and perennial reaches (Straka et al., 2019). However, this definition is region-
specific; in Mediterranean catchments, the critical period may be longer, while in Finland, it may be shorter due to differing





435 hydrological conditions. To account for these regional differences, this study employs the $RelInt10max$ indicator, which measures the annual maximum fraction of the river network that dries up over a 10-day period, as a metric for identifying extreme drying events. This indicator is conceptually similar to the Q10 flow metric used in hydrology, providing insight into low-flow conditions. The effectiveness of $RelInt10max$ was validated across various study sites. For example, in the Albarine River, the worst simulated historical event aligned with observed data. In the Bükkösdi River Basin, 2012 was identified as

440 the year with the most extreme drying within the 1984-2014 period. Similarly, while 2022 was a significant drought year for the Butiznica River, 2012 remains the most intense drought year. Such variations underscore the need for a region-specific definition of "extreme" events. In the Genal River, for instance, events with over 10 consecutive dry days are common and may not be considered extreme unless they affect the typically perennial main channel. The accuracy of $RelInt10max$ was further demonstrated in the Lepsämänjoki River, where the severe dry event of 2021 was accurately captured. However, in the

445 Velicka River, while the driest event in 2012 was correctly identified, future projections did not match this severity, suggesting potential underestimation of future extreme events under changing climate conditions. These findings highlight the importance of considering local hydrological characteristics when applying indicators like $RelInt10max$ to identify extreme events. The variability across regions emphasizes the need for a tailored approach in defining and identifying extreme drying events in river networks with different environmental characteristics.

### 4.4 Indicator calculation

The resolution of the DEMs used to delineate the spatial reach entities influences the accuracy of reach length and connectivity calculations, which evidently has an impact on the indicators. Thus, the overall river length with a certain flow condition ($LengthD/LengthF$), Table 2) is determined by the DEM used, which varies from 10 m to 30 m resolution. In addition, smaller streams and tributaries may not be well-represented in the data, which can result in underestimations of flow intermit-

455 tence in smaller streams. The calculation of the indicator ($PatchC$, Table 2) also relies on accurate representations of network connectivity. DEM resolution also determines reach connections, which should be taken into account when using this indicator to assess e.g. habitat fragmentation.

 Moreover, the uncertainties of the hydrological and RF model development and calibration also influences the indicators calculated and was described detailed in Mimeau et al. (2024).

### 4.5 Projections of flow intermittence with a RF model

Machine learning techniques are nowadays commonly used in hydrology and can be used for flood or drought forecasting (Zennaro et al., 2021; Zounemat-Kermani et al., 2021). However, they are less widely used for future projections mainly because they cannot extrapolate outside of their training range (Hengl et al., 2018). One expected result of this study was that the hybrid flow intermittence model would predict an increase in the frequency and duration of drying events under climate

465 change scenarios in reaches where some drying was observed during the training period, but that the model would continue to simulate flowing conditions, whatever the climatic conditions, for the reaches known as perennial during the training period. Results show, on the contrary, that the flow intermittence model can simulate transitions from a perennial to an intermittent





regime in some reaches. One of the reason comes from the definition of perennial and non-perennial regimes: since perennial regime is defined as less than one day of drying per year on average, it happens that a few days of drying are observed on certain
reaches during the training period without this reach being considered as intermittent. In this case, the RF model can easily predict a change towards an intermittent regime in the future, as it has been trained with some drying observations. However results show that even with a threshold strictly equal to zero dry days per year, the model can simulate a decrease of the proportion of perennial rivers for the SSP5-8.5 scenario (Fig. 10b) in all DRNs, except for the Albarine (FR) for which almost to transition from perennial to intermittent is simulated. This difference between Albarine and the other DRNs is most likely
due to the fact that the training of the RF model was much more constrained for this DRN with the addition of many flowing condition observations in perennial reaches based on the expertise of the DRN local team (see Section S1 in supplementary material and Mimeau et al. (2024)).

We assume that the ability of the flow intermittence model to predict some drying conditions that were never observed in the river networks may be due to its hybrid structure. Firstly because the JAMS-J2000 hydrological models enables to simulate
streamflow projections under climate change scenarios and provide as input data to the RF model unprecedent hydrological conditions, and secondly because the RF model is trained on the whole river network at once (and not reach by reach) which allows to predict unprecedent drying conditions in some reaches based on the drying pattern of observed in intermittent reaches during the training period.

In all cases, a major limitation of this flow intermittence model is that the projections of flow intermittence are very sensitive
to the training dataset of the RF model. As seen in the example of the Albarine, a highly constrained model in river sections considered to be perennial during the training phase, can certainly predict changes in drying patterns on historically intermittent sections, but cannot predict changes on perennial sections. On the other hand, models that are less constrained on perennial reaches might overestimate drying in the historical period and overestimate flow regime transitioning in the future projections.

Another limitation in using a model based on a RF algorithm to predict future change in flow intermittence is that the
physical processes causing flow intermittence, such as water abstractions, are not explicitly represented in the model. This introduces uncertainty, particularly for the Genal river (ES), which is intermittent in its downstream section due to abstraction for irrigation. The future intermittence projections produced by the model therefore take into account not only the impact of climate change, but also the impact of abstractions, and it is therefore not possible to separate the two impacts and determine the model's sensitivity to climate change.

**5 Conclusions and perspectives**

This study presents some future projections of drying in river networks under climate change scenarios in six European basins. The river flow intermittence is spatially simulated in river networks at a daily time step, using a hybrid hydrological model combining a spatially distributed hydrological model and a random forest model (Mimeau et al., 2024).

Flow intermittence indicators have been produced to characterise changes in drying patterns in river networks at various
spatial scales (reach scale and river network scale) and temporal scales (drying event scale, seasonal and inter-annual scale).




The projections of these indicators in the six study basins show that flow intermittence will intensify throughout the 21$^{st}$ century, whatever the GHG scenarios, at different levels depending on the basins and the scenarios considered. Overall, the results show more intense droughts, with an increase in the maximum length of the dried-up river network during the summer months (JJA), and with an increase of the duration of drying, which start earlier at the beginning of the low-flow season (late spring, early summer) and extend later into the autumn. The projections show a spatial extension of intermittent rivers with transitions from perennial to intermittent flow regimes, particularly for the SSP3-7.0 and SSP5-8.5 scenarios. An analysis of extreme drying events shows that the historical extreme events observed over the past decade could become frequent by 2071-2100 under the SSP3-7.0 and SSP5-8.5 scenarios.

An interactive web application DRYvER-Hydro (https://dryver-hydro.sk8.inrae.fr/ last access 28/08/2024; Mimeau (2023)) was developed to help visualise the evolution of intermittence indicators under climate change in the 6 study basins. DRYvER-Hydro is a Shiny application (Chang et al., 2024) developed in R (R Core Team, 2023) which allows to visualize changes in climate (temperature, precipitation, reference evapotranspiration) and flow intermittence ($RelInt$, $RelFlow$, $PatchC$) globally over the 6 basins, as well as tracking changes in flow intermittence ($ConD$, $ConF$, $DurD$, $DurF$, $numFreDr$, $numFreRW$, $FstDrE$, $IntReg$) for each river sections. Flow intermittence indicators can be displayed for the past-present period (reconstruction simulation 1960-2021) and for the future period (projections simulations 1985-2100).

The flow intermittence indicators presented in this study were designed to answer questions related to ecological issues concerning the impact of intermittency and climate change on freshwater ecosystems. In particular, these indicators have been used in interdisciplinary studies on the impact of flow intermittence on river's GHG emissions (López-Rojo et al., 2024) and the impact of flow intermittence on the degradation of micro-plastics (Barthelemy et al., 2024). Flow intermittence indicators are also being used in other studies currently being carried out as part of the European interdisciplinary H2020-DRYvER project on the impact of change on the biodiversity and ecosystem services of intermittent rivers (Datry et al., 2021).

In order to improve the quality of the flow intermittence projections and indicators for freshwater ecology studies, a future perspective of this work is to take into account more than just two conditions (flowing or dry), particularly the pool condition which can also have a significant impact on aquatic species (Steward et al., 2022). Another perspective is to improve the flow intermittence modelling by improving further the understanding of the impact of the data used to train the RF model on the simulated future projections. For these two perspectives, more observed data of flow intermittence in drying river networks will be necessary.

*Code and data availability.* The calibrated JAMS-J2000 hydrological models for the six study catchments, the R scripts used to predict flow intermittence with a random forest algorithm, the downscaled climate projections, the observed flow state data, and the all the computed flow intermittence indicators used in this study can be obtained from the corresponding author upon request.





*Author contributions.* Conceptualisation: LM, AK; model implementation and analysis: LM, AK, AD; climate projections downscaling: AD, JPV, CL; flow intermittence indicators computation: AK, SK; draft preparation and discussions: LM, AK, FB, SK, JPV. Local DRN observations and expertise: NB, ZC, DT, HM, PP, LP; All authors read and approved the final paper.

*Competing interests.* The contact author has declared that none of the authors has any competing interests

*Acknowledgements.* This study was supported by the H2020 European Research and Innovation action Grant Agreement no. 869226 (DRYvER).



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
