# Peer review of "Projections of streamflow intermittence under climate change in European drying river networks"

_Hydrology and Earth System Sciences, 2024_

## Author Response (AR1)

**Response to Editor's Comments**

Dear Khalid Hassaballah

Thank you for the constructive feedback and careful evaluation of our manuscript. Below, we provide detailed responses to each of the points raised and outline the changes made in the revised manuscript to address them.

1. Why is the use of a hybrid model necessary to predict flow intermittence, and what justifies developing a new model despite the availability of numerous existing hydrological models?

We have elaborated on the justification for employing a hybrid model to predict flow intermittence. Specifically, we added the following paragraph in section 2.2 "Flow intermittence model":

There are several reasons for using a hybrid model in this study rather than a model, such as ParFlow, HGS, or MIKESHE, which are able to represent groundwater-river interactions and directly simulate the drying up of rivers. A first reason is that these physically based models are limited by high data requirements, computational costs, and difficulties simulating zero-flow conditions, which are critical for predicting flow intermittence (Shanafield et al., 2021; Gutierrez-Jurado et al., 2021; Herzog et al., 2021). Additionally, the origins of flow intermittence are complex, involving local geological, climatic, and anthropogenic processes, making it difficult for physically-based models to capture these dynamics without extensive datasets (Datry et al., 2016,2018). The hybrid modelling technique used in used in this study, integrate physical hydrological modeling and machine learning to enable robust long-term projections of flow intermittence across high spatial and temporal scales, minimizing data requirements and consequently high computational efforts (Mimeau et al., 2024).

2. Data Scarcity and Model Validation: Provide a more comprehensive discussion on how data limitations were addressed and the steps taken to validate the model given these constraints.

As requested by Referee RC2, a table (Table 2) providing the number of reaches per DRNs as well as the total length of the river network was added in section "2.1 Study sites".

This issue of data scarcity and model validation has already been fully described and discussed in our previous publication (Mimeau et al., 2024) and complementary data showing the data availability in the DRNs was already provided in supplementary material. Given the already substantial content of this manuscript, we have preferred not to discuss this issue again in this paper and to discuss in greater depth the uncertainties associated with applying the model to future projections with climate change scenarios.

We added the following sentence to make it clearer that this issues had already been addressed: "The uncertainty related to the training of the RF models under data scarcity is discussed in greater details in Mimeau et al., 2024)." on Line 179.

3. Justification for Site Selection: Offer a clearer and more detailed explanation of the criteria used to select the study sites and how these choices impact the study's broader applicability.

We added another subsection to the discussion to address this topic: "4.6 Spatial patterns of flow intermittence changes at the European scale".

«The 6 European DRNs studied in the DRYvER project, were selected to evaluate the impact of climate change on flow intermittence in different regions characterized by diverse climatic and hydrological regimes across Europe. Although the 6 study basins are local basins of relatively small size, their locations can provide information about the impacts of climate change on intermittency on a larger scale on the European continent. The reconstruction simulations (Fig. 1) show the differences in intermittence between the different climatic regions: the DRNs in southern Europe with dry climates (Genal, Butiznica) are very intermittent, whereas the DRN in northern Europe (Lepsämänjoki) is relatively little intermittent. The flow intermittence projections under climate change scenarios also shows a general tendency for the percentage of intermittent river network to increase more in the southern sites than in the north (Fig. 6,7,8). However, our results also show that local processes, such as geological characteristics and water abstractions, have a significant influence on intermittence. This suggests that the climatic region alone is insufficient to fully explain intermittence at the scale of local river networks.»

    4. Geometry of River Networks: Include a more in-depth description of how river network geometry was accounted for in the modelling process and its implications for the results.

The role of river network geometry is now discussed more detailed in subsection "4.4 Influence of the river geometry on the flow intermittence indicators", which was formerly called "Indicator calculation". We added: "The river network geometry influences hydrological processes by shaping flow pathways, connectivity, and retention across the network (Roy et al., 2022). Properties such as number of reaches, sinuosity and hydrological connectivity, directly affect flow dynamics, including flow intermittence. For example, steeper upstream reaches typically lead to faster runoff and shorter flow durations, while flatter downstream reaches can sustain pooling or prolonged drying phases, as observed in our case studies. Future studies could benefit from incorporating additional metrics and methodologies, such as Horton's Laws or advanced clustering approaches (Roy et al., 2022) to further investigate the influence of river network geometry on hydrological processes. The limited number of six case studies constrains the generalization of these findings, but the results highlight the significant role of network geometry and connectivity in shaping flow intermittence and seasonality.»

    5. Model Accuracy Assessment: Expand on the methods used to assess the model's accuracy and reliability, ensuring that the evaluation process is transparent and robust.

Referee RC2 asked to discuss more about the attribution efficiency to variations of the model structure. This issue was addressed in the new section "Influence of the river geometry on the flow intermittence indicators" as stated above.

New references cited in our manuscript have been incorporated into the revised manuscript. A full list of references has been updated accordingly.

We hope that these revisions adequately address your concerns and enhance the clarity of the manuscript. We appreciate the opportunity to revise our work and look forward to your feedback.

Sincerely,

Louise Mimeau on behalf of the co-authors

References:

Gutierrez-Jurado, K. Y., Partington, D., and Shanafield, M.: Taking theory to the field: streamflow generation mechanisms in an intermittent Mediterranean catchment, Hydrol. Earth Syst. Sci., 25, 4299–4317, https://doi.org/10.5194/hess-25-4299-2021, 2021.

Herzog, A., Hector, B., Cohard, J. M., Vouillamoz, J. M., Lawson, F. M. A., Peugeot, C., and de Graaf, I.: A parametric sensitivity analysis for prioritizing regolith knowledge needs for modeling water transfers in the West African critical zone, Vadose Zone Journal, 20(6), e20163, https://doi.org/10.1002/vzj2.20163, 2021.

Mimeau L, Künne A, Branger F, Kralisch S, Devers A, Vidal J-P (2024) Flow intermittence prediction using a hybrid hydrological modelling approach: influence of observed intermittence data on the training of a random forest model. Hydrology and Earth System Sciences. https://doi.org/10.5194/hess-28-851-2024

Roy, J., Tejedor, A., & Singh, A. Dynamic Clusters to Infer Topologic Controls on Environmental Transport of River Networks. Geophysical Research Letters, 49, 1–11. https://doi.org/10.1029/2021GL096957, 2022

Shanafield, M., Bourke, S. A., Zimmer, M. A., and Costigan, K. H.: An overview of the hydrology of non-perennial rivers and streams, Wiley725Interdisciplinary Reviews: Water, 8, e1504, https://doi.org/10.1002/wat2.1504, 2021.